# Rethinking Mutual Information for Language Conditioned Skill Discovery on Imitation Learning

**Primary Keywords:** *Learning, Imitation Learning, Information Theory*

## Abstract

Language-conditioned robot behavior plays a vital role in executing complex tasks by associating human commands or instructions with perception and actions. The ability to compose long-horizon tasks based on unconstrained language instructions necessitates the acquisition of a diverse set of general-purpose skills. However, acquiring inherent primitive skills in a coupled and long-horizon environment without external rewards or human supervision presents significant challenges. In this paper, we evaluate the relationship between skills and language instructions from a mathematical perspective, employing two forms of mutual information within the framework of language-conditioned policy learning. To maximize the mutual information between language and skills in an unsupervised manner, we propose an end-to-end imitation learning approach known as Language Conditioned Skill Discovery (LCSD). Specifically, we utilize vector quantization to learn discrete latent skills and leverage skill sequences of trajectories to reconstruct high-level semantic instructions. Through extensive experiments on language-conditioned robotic navigation and manipulation tasks, encompassing BabyAI, LORel, and Calvin, we demonstrate the superiority of our method over prior works. Our approach exhibits enhanced generalization capabilities towards unseen tasks, improved skill interpretability, and notably higher rates of task completion success.

## Introduction

General-purpose robots operating alongside humans in their environment must develop the ability to understand and respond to human language in order to perform a wide range of complex tasks. Currently, there is significant research interest in language-conditioned policy learning methods, such as Vision-Language Navigation (VLN) (Gu et al. 2022) and Vision-Language Manipulation (VLM) (Guhur et al. 2023; Shridhar, Manuelli, and Fox 2023), which aim to enable robots to learn the connection between language instructions and their perceptions and actions.

In multi-task scenarios, tasks are typically defined by different task IDs (Gupta et al. 2019; Yu et al. 2020). However, in complex environments, task IDs do not capture the relationships between tasks effectively and can be labor-intensive to define. On the other hand, human language provides a more natural and flexible way to define and specify tasks. Additionally, robots need to acquire a diverse set of general-purpose skills that enable them to understand unconstrained language instructions and perform long-horizon tasks.

Most modern skill-learning methods are limited to task ID settings and sparse reward reinforcement learning (RL) environments. Hierarchical reinforcement learning (HRL) approaches to address complex tasks by learning latent skills, which are then used in low-level meta-control (Haarnoja et al. 2018). Other approaches decouple skill state mutual information into forward (Sharma et al. 2019; Campos et al. 2020; Laskin et al. 2022) and reverse (Gregor, Rezende, and Wierstra 2016; Eysenbach et al. 2018; Achiam et al. 2018) forms, which are incorporated into the reward function. These works offer theoretical analysis and outperform other methods in RL benchmarks(Todorov, Erez, and Tassa 2012). However, these approaches have not been applied to language-conditioned policies.

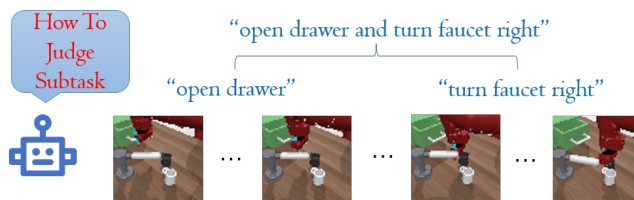

Figure 1: An example of multi-task language conditioned situation.

As depicted in Figure 1, given a task specification like *open drawer and turn faucet right*, traditional language-conditioned policy struggles to effectively differentiate the subtasks contained within language instructions based on different states(Guhur et al. 2023). Contrastive learning is commonly employed for establishing multimodal relationships(Eysenbach et al. 2022). However, this approach typically requires pre-labeling of corresponding image sequences and language subtasks, which can hinder generalization. By learning discrete skills, we can fully demonstrate the generalization ability of our imitation model in multi-task scenarios without refining tasks.

Mapping complex languages to discrete skill spaces presents a challenge. In this paper, we experimentally found that skills can directly relate to language instructions, al-

lowing for direct optimization based on their mutual relation. Moreover, in multi-task language-conditioned environments, as illustrated in Figure 1, latent skills specified in language instructions need to be constrained by the state.

To address these challenges, we propose the Language Conditioned Skill Discovery (LCSD) method to tackle the imitation learning problem in multi-task environments. Our approach is based on mutual information theory, which establishes the relationship between discrete skills, the current state, and language instructions. We employ the VQ-VAE method for skill learning, where the encoder decomposes language and the current state while the decoder aims to reconstruct unique discrete skills and convert them back into language. To generate diverse skills, we introduce code reinitialization to prevent index collapse. We utilize the diffusion policy with the U-net denoising model as an imitation policy, which exhibits better adaptability to different environments.

We conduct experiments in robotic manipulation and 2D navigation to evaluate the effectiveness of LCSD. Compared with language condition policies and skill-based imitation models, our method outperforms prior works. LCSD demonstrates superior generalization, skill interpretability, and task completion rates. Notably, it achieves a 20% improvement in complex robot manipulation tasks.

To summarize, our contributions are as follows:

- We propose a skill-learning method based on mutual information that establishes the relationship between state, skill, and language.

- We introduce LCSD, a hierarchical skill learning Imitation policy based on VQ-VAE and diffusion model for long-horizon, language-conditioned multi-task environments.

- We show that our skill discovery method provides better interpretable discrete skills in different environmental conditions than previous methods.

- We demonstrate that our method outperforms existing methods in language-conditioned multi-task environments.

## Related Work

### Language Conditioned Policy

Prior research has primarily addressed decision-making in complex tasks that involve language instructions, particularly in robot environments (Shridhar, Manuelli, and Fox 2022; Nair et al. 2021). Existing work has focused on employing pre-trained language models (Radford et al. 2021; Devlin et al. 2018; Chowdhery et al. 2022) as language encoders due to the complexity and diversity of human languages. Some previous studies have used behavior cloning to align the output of pre-trained language models with observation inputs in order to predict actions (Shridhar, Manuelli, and Fox 2022; Zheng et al. 2022). Other approaches have explored LLM (Language Model)-based prompt engineering to decompose complex language instructions into sub-tasks (Brown et al. 2020; Ahn et al. 2022). A closely related work to ours is Saycan (Ahn et al.

2022), as both our work and Saycan aim to generalize latent skills using languages and states. However, Saycan requires a pre-defined set of skills to estimate the Q-function for each skill, whereas we can extend our skills to unknown tasks by utilizing a codebook of varying sizes.

### Skill Discovery via mutual information

Skill discovery has been primarily employed in Hierarchical Reinforcement Learning (HRL). Agents select latent variables from a set of skills at the high-level policy, which is then executed by a meta-controller to perform sub-tasks (Haarnoja et al. 2018; Shi, Lim, and Lee 2022). Recent studies have emphasized encouraging agents to explore and have often relied on the mutual information between states and skills (Gregor, Rezende, and Wierstra 2016; Campos et al. 2020). However, few works have addressed skill learning in a language-conditioned environment. LISA (Garg et al. 2022) utilizes a skill predictor based on states and language within specific horizons. Nevertheless, a single encoder cannot establish a direct connection between skills and language, leading to instability in skill learning.

## Preliminary

**Mutual Information Skill learning:** Mutual Information(MI) is a measure of the statistical dependence between two variables. Given state $s$ and skill $z$, the mutual information $I(z; s)$ can be optimized in two ways(Campos et al. 2020). The forward form: $I(z; s) = H(s) - H(s|z)$, where $p(s|z)$ is estimated by a variational approximation, state entropy is approximated by expectations of $p(s|z)$ estimated over all skills(Campos et al. 2020; Sharma et al. 2019; Park et al. 2023). In the reverse form $I(z; s) = H(z) - H(z|s)$, latent code $z$ is sampled from a fixed distribution and the lower bound of conditioned entropy is estimated by $\rho_\pi(z|s)$(Eysenbach et al. 2018; Gregor, Rezende, and Wierstra 2016).

**VQ-VAE:** Vector Quantized Variational Autoencoder (VQ-VAE)(Van Den Oord, Vinyals et al. 2017) is a neural network architecture for unsupervised learning of latent representations of data. In VQ-VAE, the encoder maps the input data to a continuous latent space, which is then quantized to a discrete codebook. The decoder maps the discrete code to the output space, generating new samples. VQ-VAE updates the encoder, decoder, and codebook parameters with the following loss function.

$$L = \log p\left(x \mid q(z_q^k)\right) + \left\| sg\left[p(x)\right] - z_q^k \right\|_2^2 + \beta \left\| p(x) - sg[z_q^k] \right\|_2^2 \tag{1}$$

the first term represents the reconstruction from discrete code to original input for updating the encoder $p$ and decoder $q$. The second term leads the discrete vectors in the codebook $z_q^{1\cdots N}$ to approach the nearest output of the encoder, while the last term is commitment loss, encouraging the output of the encoder to stay close to the chosen kth codebook vector $z_q^k$. The codebook update can also use exponential moving averages instead of the second in the loss function.

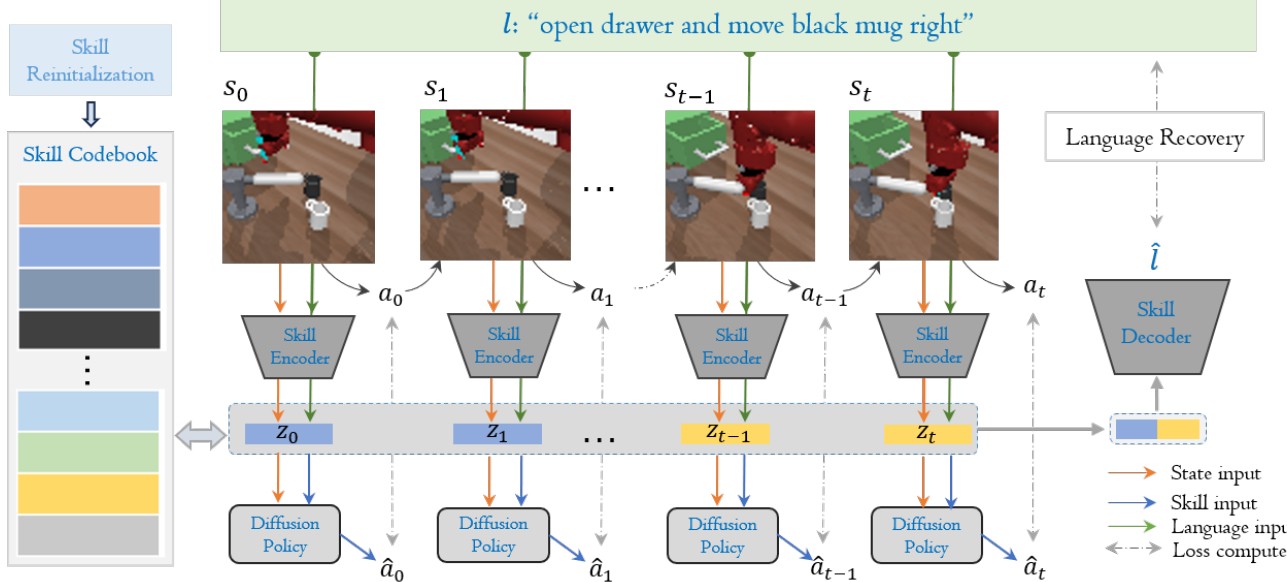

Figure 2: **Overview of LCSD**. In the skill learning stage, the encoder decomposes the current state and language to a lower-dimensional latent space, while the decoder recovers the quantized latent skills to the language embeddings. A single vector is chosen from the codebook in each step and used to quantize the encoder outputs. The diffusion model is used as an action predictor conditioning on current state and skill(or language).

## Approach

LCSD is a two-stage imitation learning structure that comprises an encoder-decoder model for skill acquisition and a conditional diffusion policy for action prediction. In the first stage, the skill encoder and decoder learn a codebook of latent skill vectors corresponding to languages conditioned on states. The diffusion policy then predicts the subsequent action directly, conditioned on the current state and latent skill generated by the skill encoder. An overview of our approach is depicted in Figure 2.

### Problem Formulation

We consider learning in general environments modeled as the Markov decision processes (MDPs). In each environment, we are provided with an offline dataset consisting of $N$ demonstration sequences obtained from a diverse set of tasks using a behavior policy. Each trajectory consists of state-action pairs with one language label over $T$ time steps.

For multi-task environments, each language describes a set of tasks with varying quantities. The states and actions performed by the agent were stored as pairs along with a single language instruction in each trajectory .

$$\tau_i = \{s_0, a_1, a_1, ..., s_T, a_T, l\}_{i=0}^{N}.$$

### Mutual Information Skill Learning in LCSD

In language-based imitation learning environments, the agent executes actions based on tasks specified through language. Therefore, the skills we learned must closely relate to the language instructions. Firstly, we directly maximize the mutual information between skills and language

$I(\mathbf{z}, l)$, where $\mathbf{z}$ represents skill sets for the entire trajectory. In multi-task environments, a single language may involve multiple skills, as shown in Figure 1. In such cases, skills need to segment the trajectory into sub-tasks based on different states. For example, when executing the instruction *open the drawer and pick up the cup*, our skill needs to distinguish the current task of the agent based on whether the drawer is already open or not. To this end, we further aim to maximize the mutual information between skill and language conditioned on the current state, denoted as $I(l; z|s)$. In summary, our goal is to maximize:

$$
\begin{aligned}
\mathcal{F} &= I(\mathbf{z}; l) + I(l; z|s) \\
&= H(l) - H(l|\mathbf{z}) + H(z|s) - H(z|l, s) \quad (2) \\
&= H(z|s) + E_{z \sim p, s \sim D}[\log p(z|s, l)] + \\
& \quad E_{z \sim p, l \sim D}[\log p(l|\mathbf{z})] + \text{Const}, \quad (3)
\end{aligned}
$$

As shown in Equation 2, we use forward form on $I(\mathbf{z}; l)$ and reverse form on $I(z|l, s)$. $H(l)$ represents the entropy of language instructions, which is constant in our offline dataset. The second term focuses on how our skills are related to language. The third term expects our skill distribution to have high entropy for better generalization conditioned on states. For the last term, our goal is to map deterministic discrete skills with current state and language instructions as conditions.

In Equation 3, we express the formula in the form of a probability distribution, where skills are sampled from a uniform distribution $p(z)$, and states $s$ and language $l$ are sampled from the stationary offline dataset. We implicitly optimize $H(z|s)$ by initializing unused codes for a broader range of skill selection and explicitly approximate the lower bound

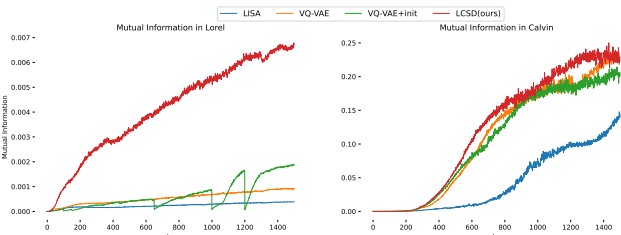

Figure 3: **MI training curve in Calvin and LORel with difference skill learning methods**. We show the mutual information curves of our method during training in different environments on different skill learning methods.

of conditional probability distribution by neural networks, the skill encoder $p_\theta(z|l,s)$ and the skill decoder $q_\phi(l|\mathbf{z})$. The encoder constrains the predicted skills for each step, while the decoder updates macroscopic language instruction reconstruction after skill generation on the entire trajectory. The ultimate optimization goal can be simplified as maximizing the lower bound of our objective $\mathcal{F}(\theta,\phi)$:

$$\mathcal{F}(\theta,\phi) \geq H(z|s) + E[\log p_\theta(z|s,l)] + E[\log q_\phi(l|\mathbf{z})]. \quad (4)$$

We plot MI curves in four skill acquisition methods in Figure 3: skill learning method in LISA(Garg et al. 2022), which only contains a skill encoder, LISA with code reinitialization(to maximize $H(z|s)$), VQ-VAE(last two terms on Equation 4), and LCSD. The metric value represents the mutual information between the skill and language-state, $I(z;l,s) = H(l,s) - H(l,s|z)$. Figure 3 demonstrates that the mutual information increases more significantly when the skill-language decoder and code reinitialization are utilized. Compared to the Calvin environment, a single trajectory corresponds to multiple sub-tasks in the LORel dataset, which requires a stronger correlation between language and skills. Hence, adding a decoder proves more effective in improving MI in the LORel environment compared to Calvin. More details of the skill learning structure are shown below.

## Skill learning

VQ-VAE is an unsupervised generative model for representation learning that uses an encoder to map images into latent space and a decoder to reconstruct the original image. In previous works on imitation learning, a skill encoder was used to directly map states to skills without a decoder, resulting in unstable, non-interpretable skills for task analysis (Sudhakaran and Risi 2023; Garg et al. 2022). (Mazzaglia et al. 2022) used the complete VQ-VAE framework for skill discovery, where a decoder was used to reconstruct states for computing rewards to update the world model in Actor-Critic training. LISA (Garg et al. 2022) introduced language into VQ training to solve decision-making problems with IL. However, a single encoder mapping discrete skills from language-state embeddings is inadequate in learning the direct relation between skills and languages, resulting in poor stability in different environment settings(Figure 6). More LISA skill maps are shown in the Appendix.

To address this, we jointly map the state and language to latent skill vectors when selecting skills, following VQ-VAE. The skill encoder $p_\theta(s,l)$ learns as a language-state

representation. In vector quantization (VQ), a codebook comprising $M$ latent codes of skill vectors $z^{1\ldots M}$ is utilized, and the skill vector closest to the encoder output is selected.

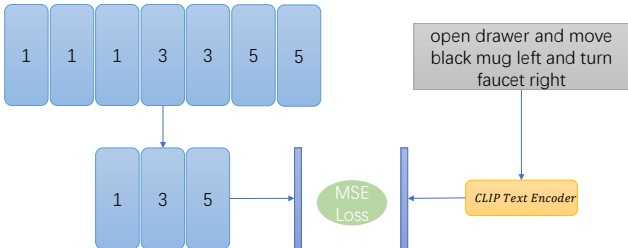

Figure 4: **Instruction Semantic Recovery Diagram**. The decoder's objective is to choose a distinct skill from each consecutive group within a trajectory and calculate the mean squared error (MSE) loss using the frozen CLIP language embedding.

**Instruction Semantic Recovery:** In language-based multi-task environments, we propose a language reconstruction-based VQ-VAE to better learn the specific tasks corresponding to skills in different states. We introduce a decoder corresponding to the last term of our optimization goal (Equation 4), which solely aims to align the skill vectors with the language representation. Unlike previous work(Mazzaglia et al. 2022), our decoder does not participate in subsequent policy updates but rather serves to optimize the correspondence between the embeddings generated by the skill encoder and the language.

We present the structure of our skill decoder in Figure 4. As language contains varying amounts and types of skills in different trajectories, we consider consecutive selections of the same skill as a sub-task and choose the first code from every consecutive group of equivalent skill codes. The decoder takes a discrete set of skills as input and outputs a vector, which is compared to the language vector obtained from a frozen CLIP text encoder using MSE loss.

$$\begin{aligned} L_{skill} &= -\log q_\phi(l|\mathbf{z}) - \log p_\theta(z|s,l) \\ &= \|q_\phi(U(z_q(s,l))) - E(l)\|_2^2 + \quad (5) \\ &\quad \beta \|p_\theta(s,l) - sg(z_q)\|_2^2. \end{aligned}$$

$U$ represents the unique selection of discrete skills, and $E$ refers to the CLIP text encoder(Radford et al. 2021) that is frozen during training. The skill loss is comprised of two components: the reconstruction loss and the commitment loss, both of which are incorporated in the VQ training procedure according to Equation 1. By minimizing the skill loss, we simultaneously maximize the Mutual Information based on Equation 4.

**Skill Reinitialization:** Selecting a single code may lead to index collapse during the skill learning period due to preferential selection (Kaiser et al. 2018). We also encountered similar situations in some environments during evaluation, as shown in the upper image in Figure 6. To address this issue, we used codebook reinitialization, which aims to involve more codebook vectors in skill learning and code selection. Inspired by (Mazzaglia et al. 2022), We recorded

times of codes selected within a certain training iteration and reinitialized codes with proper skill encoder vectors. Reinitialized codes can directly participate in skill selection and updates in subsequent training.

We selected encoder outputs $p_\theta(s_i, l)$ to reinitialize inaccessible code $z_q^k$ with a probability of $\frac{d_p^2(z_q^k, s_i, l)}{\sum_s d_p^2(z_q^k, s_i, l)}$, Where $i$ is the index corresponding to the output of the encoder that selected the replacement code $p_\theta(s, l)$ is the output from the skill encoder on the last training batch, $N$ represent the total number of skills in the codebook, and $d_p$ is the fraction of the Euclidean distance between the encoder output and codebook embedding. Unlike (Mazzaglia et al. 2022), we choose an embedding closer to the current code for more stable skill generalization.

$$d_p^2(z_q^k, s_i, l) = \frac{1}{\left\| p_\theta(s_i, l) - z_q^k \right\|_2^2}. \tag{6}$$

Unlike previous works, our approach incorporates a generalized strategy of not only initializing unused codes but also resetting the entire code set with a specified probability. This decision is motivated by our observation that inactive skills during the initial training phase can lead to significant instability in the early stages of training. To determine the reset probability, we calculate it proportionally to the ratio of each skill to the average skill selection. By adopting this method, we aim to enhance the stability and effectiveness of the training process.

$$z_q^k = \begin{cases} p(\frac{d_p^2(p_\theta(s_i, l))}{\sum_s d_p^2(p_\theta(s, l))}) * p_\theta(s_i, l), & y > \frac{M_k * N}{\sum_{j=1}^N M_j}, \\ z_q^k, & y < \frac{M_k * N}{\sum_{j=1}^N M_j} \end{cases} \tag{7}$$

Where $M$ represents the number of times each skill is selected during this training session, and $y$ is a randomly generated float value ranging from 0 to 1. We aim to initialize the code with fewer prior selections, enabling more efficient updates in subsequent training iterations. We anticipate improved training efficiency and effectiveness by prioritizing initializing less frequently selected codes. Notably, our skill reinitialization method only takes place in the first 200 epochs of training, as we aim to make the most of each skill as possible while maintaining the way skills are learned. Therefore, fewer steps to initialize can avoid excessive human intervention in training and achieve better results.

## Diffusion policy for Imitation Learning

We adopt the Denoising Diffusion Probabilistic Model (DDPM) (Ho, Jain, and Abbeel 2020) as our policy base model. The denoising network aims to predict the random noise added to the action in each iteration. The noisy input in each iteration can be formulated as $\mathbf{a}_i = \sqrt{\bar{\alpha}_i}\mathbf{a} + \sqrt{1 - \bar{\alpha}_i}\boldsymbol{\epsilon}$, where $\bar{\alpha}$ are process variances, and random noise $\boldsymbol{\epsilon}$ is sampled from a Gaussian distribution $N(0, I)$. As an imitation policy to solve language condition tasks, our diffusion model can support language or skill information along with the current state as input simply by modifying the conditional input

dimension. We modified the policy training loss as follows:

$$\mathcal{L}_{ddpm-s}(\theta) = E_{\boldsymbol{\epsilon}, i, s, a, l, \boldsymbol{z}} \left[ \left\| \boldsymbol{\epsilon} - \boldsymbol{\epsilon}_\theta \left( \mathbf{a}_i, \boldsymbol{s}, \boldsymbol{z} \right) \right\|^2 \right].$$
$$\text{s.t.} \quad \boldsymbol{\epsilon}, i \sim \mathcal{U}, (s, a, l) \sim \mathcal{D}, \boldsymbol{z} \sim z_q(s, l) \tag{8}$$

Where $i$ is sampled from $\mathcal{U}[1, T]$, denoise network $\epsilon_\theta$ is trained to predict random noise with state, action noise, and skill(or language) as input.

To combine skill (or language) and image features, we used different linear layers similar to the Temporal U-Net as our diffusion denoising network. For each MLP block, separated linear layers were used to unify the dimensions of the action noise, state, timestep, and skill embedding (language), and then they were added together. The final linear layer of the network outputs noise with the same dimension as the action. This network was designed to fully utilize conditional information. The detailed structure is shown in the Appendix.

LCSD is an end-to-end imitation policy, and we provided an overall structure feature in Figure 2. We developed a high-level skill generator based on a VQ-VAE model, which discretizes the latent space. The generated skills were then used in a diffusion policy as conditional information to predict the next-step action. The overall loss combines skill and imitation policy as:

$$\mathcal{L}_{LCSD} = \alpha \mathcal{L}_{skill} + \gamma \mathcal{L}_{ddpm-s},$$

where $\alpha$ and $\gamma$ are used to balance the behavior cloning (BC) and skill learning losses.

---

**Algorithm 1: LCSD**

---

Initialized Model: diffusion policy $\pi$, skill encoder $p_\theta$, skill decoder $q_\phi$, CLIP encoder $\mathcal{E}$, Codebook quantize on encoder q

**for** *training iterations $i = 1...N$* **do**
    Sample batch $\tau = \{l, (s_0, a_0), (s_1, a_1), ...(s_T, a_T)\}_{i=0}^B$
    *Skill learning Period*
    **if** *Skill learning* **then**
        **for** *each trajectory $\tau$* **do**
            $z_{0:T} \leftarrow p_\theta(s_{0:T}, \mathcal{E}(l))$
            record unselected codes in list $u$
        **end**
        Compute skill loss $\mathcal{L}_{skill}$ with Equation $\mathcal{L}_{skill} = \mathcal{L}_{reconstruct} + \mathcal{L}_{commitment}$
        **if** $i < reinitupdate$ *and* $i \bmod reinitstep = 0$ **then**
            reinitialize unused code in list $u$ with probability on Equation 7.
        **end**
    **end**
    *Behavior Cloning Period*
    **if** *Skill learning* **then**
        $a'_{0:T} = \pi(s_{0:T}, q(p_\theta(s_{0:T}, \mathcal{E}(l))))$
        **else**
            $a'_{0:T} \leftarrow \pi(s_{0:T}, \mathcal{E}(l))$
        **end**
        Compute Behavior cloning loss $\mathcal{L}_{ddpm-s}$
        update with $\mathcal{L}_{LCSD} = \alpha \mathcal{L}_{skill} + \gamma \mathcal{L}_{ddpm-s}$
    **end**

---

Table 1: **Success rate for all tasks**. We show our LCSD performance in different environments compared to the Baselines mentioned below. The best method is shown in bold.

| Task | Original | Lang+DT | LISA | LISA+init | Lang+Diffusion | LCSD |
|------|----------|---------|------|-----------|----------------|------|
| BabyAI GoToSeq | $40.4 \pm 1.2$ | $62.1 \pm 1.2$ | $65.4 \pm 1.6$ | - | $65.2 \pm 8.6$ | $\mathbf{67.8 \pm 8.2}$ |
| BabyAI SynthSeq | $32.6 \pm 2.5$ | $52.1 \pm 0.5$ | $53.3 \pm 0.7$ | - | $55.1 \pm 2.5$ | $\mathbf{57.6 \pm 2.2}$ |
| BabyAI BossLevel | $28.9 \pm 1.3$ | $60.1 \pm 5.5$ | $58.0 \pm 4.1$ | - | $55.0 \pm 3.4$ | $\mathbf{60.5 \pm 7.4}$ |
| LORel sawyer state | $6 \pm 1.2$ | $33.3 \pm 5.6$ | $6.7 \pm 3.3^*$ | $43.4 \pm 0.2$ | $43.0 \pm 1.5$ | $\mathbf{60.2 \pm 5.7}$ |
| LORel sawyer obs | $29.5 \pm 0.07$ | $15.0 \pm 3.4$ | $10.3 \pm 1.4^*$ | $24.5 \pm 4.3$ | $36.6 \pm 3.8$ | $\mathbf{45.5 \pm 5.1}$ |
| Calvin | $32.5 \pm 2.5$ | $11.7 \pm 0.8$ | $10.1 \pm 3.3$ | $10.9 \pm 0.4$ | $\mathbf{37.5 \pm 2.6}$ | $33.6 \pm 1.3$ |

$^*$ We optimize LISA with official code from (Garg et al. 2022) but cannot get normal performance on LORel due to index collapse.

## Experiments

### Tasks

To verify the LCSD's effectiveness, we selected three benchmarks: LORel Sawyer dataset (Nair et al. 2021), BabyAI navigation (Chevalier-Boisvert et al. 2018), and Calvin robot tasks (Mees et al. 2022), which are all language-based and imitation learning environments without reward. Other benchmarks either lack language conditioning settings (Yu et al. 2020; Gupta et al. 2019) or focus on single-task environments with complex observation representations that generate hardly interpretable skills (Shridhar, Manuelli, and Fox 2022).

For the BabyAI benchmark, we used 10k trajectories evaluating three challenging tasks, namely GoToSeq, SynthSeq, and BossLevel. We collected an offline dataset of 50k trajectories on LORel and evaluated the performance on several task settings. For the Calvin benchmark, we directly select 1216 trajectories from the Calvin-D dataset relevant to the six tasks we modified. To eliminate interference on image encoders and focus solely on evaluating the underlying policy, we directly select the 21-dimensional perspective state of the Calvin environment as observation input. More information on datasets is shown in the Appendix.

### Baselines

We compared our proposed LCSD with several baselines in our experiments:

**Original**: The BC baselines from original papers on three benchmarks. In BabyAI we adopt their RNN-based method(Chevalier-Boisvert et al. 2018). In the LORel environment, we compared with the planner algorithm as language conditioned BC baseline. We trained MULC from (Mees et al. 2022) on our Calvin setting by changing the vision encoder into a simple MLP for perspective state observation.

**Language conditioned DT policy**: A behavior cloning Decision Transformer(DT)(Chen et al. 2021) based policy that takes the language instruction and past observations as inputs to predict action.

**LISA**(Garg et al. 2022): A hierarchical imitation learning structure based on a skill encoder and DT based policy.

**LISA with code reinitialization**: LISA with code reinitialize to better generalize skill code, denoted as LISA init.

**LCSD on DT policy**: We apply the skill learning method of LCSD to the DT policy, referred as LCSD+DT.

**Language Condition Diffusion Policy**: An imitation learning structure with diffusion policy condition on language. Different from LCSD, the input of the diffusion model is language tokens generated by the CLIP text encoder and current observation. We modify the structure by directly minimizing the behavior cloning loss in Equation 9.

$$\mathcal{L}_{ddpm-l}(\theta) = E_{\epsilon, i \sim \mathcal{U}, (s,a,l) \sim \mathcal{D}} \left[ \| \epsilon - \epsilon_\theta \left( \mathbf{a}_i, s, l \right) \|^2 \right]. \tag{9}$$

### Results

We evaluated our approach in three environments. BabyAI serves as the most straightforward task with discrete actions for 2D navigation, while LORel is a medium-difficulty multi-task language environment based on Metaworld. Table 1 presents the overall results for different tasks. All algorithms were trained for 1500 iterations over three seeds. Notably, LCSD outperformed the other language condition BC methods in various tasks. Specifically, LCSD showed superior performance in multi-task and complex LORel environments.

**Diffusion model can leverage stability in difficult tasks:** While serving as a long-horizon benchmark, the language label in Calvin corresponds to a single skill, which is different from the other two benchmarks (See appendix for more dataset details). We mainly introduce Calvin to evaluate the performance of different imitation learning models on difficult tasks rather than to measure the effect of skill learning. The dataset we selected for Calvin only contains only 1216 trajectories, making the tasks even more challenging.

Our diffusion policy performed well in different tasks without requiring special modifications to the parameters, particularly excelling four times in Calvin tasks(Comparing DT and Diffusion column in Table 1). Table 2 lists the success rates in six different tasks, clearly showing that the diffusion-based policy outperforms DT-based models. It is typical for language-based models to exhibit slightly superior performance compared to skill-based models as a result of employing limited training data in Calvin, along with massive redundant data. More detailed information about Calvin's task settings can be found in the Appendix.

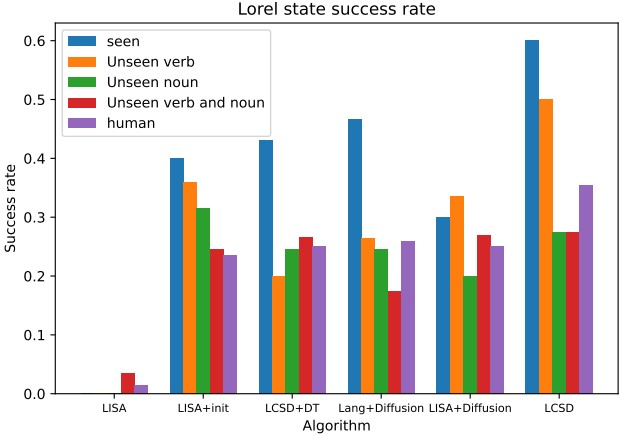
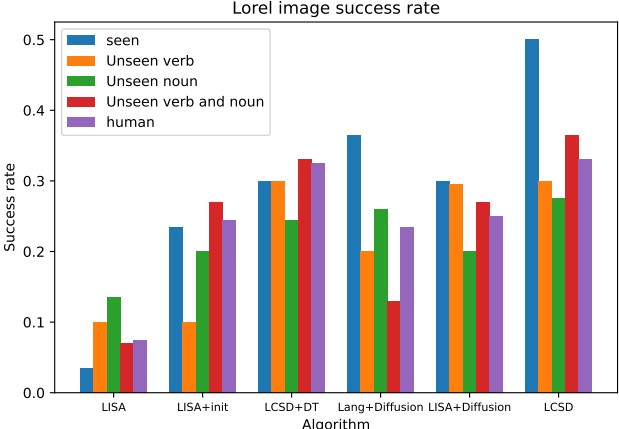

Figure 5: **The overall task success rate in LORel Sawyer Environment**. Performance of different algorithms on different task settings on LORel Sawyer state and image environments. The three algorithms on the left are all based on the DT model, while the three on the right are based on the diffusion model.

Table 2: Success rate on Calvin tasks

| Calvin Tasks | DT model | | Diffusion model | |
|---|---|---|---|---|
| | language | skill | language | skill |
| Turn on ledbulb | 0 | 0 | 0.63 | 0.13 |
| Turn off ledbulb | 0.25 | 0.13 | 0.25 | 0.13 |
| Move slider left | 0 | 0.13 | 0.12 | 0.38 |
| Move slider right | 0 | 0 | 0.5 | 1.0 |
| Open drawer | 0 | 0 | 0.25 | 0.13 |
| Close drawer | 0.25 | 0.25 | 0.5 | 0.25 |
| Overall | 0.083 | 0.085 | **0.375** | 0.336 |

**Skill Visualization:** To demonstrate the specific meaning of the discrete skills generated by our algorithm, we recorded the correlation between language and the selection of skill codebook during evaluation, as done in (Garg et al. 2022). We first show the skill map of (Garg et al. 2022) in the upper image and find that most of the skills in the codebook are not involved in training. Different tasks can only be divided into two types of skills, which cannot be effectively trained, lea-ding to index collapse. The lower image in Figure 6 shows that all 20 codes were selected during evaluation, with a strong correspondence observed between language tokens and skill codes. For instance, Skill code 19 (the nineteenth column) corresponds to the action "*turn/rotate faucet right/clockwise*," while skill code 0 (the zero column) represents "*rotate handle rightward and open drawer*." It is reasonable for a single skill to indicate these two tasks because in LORel, the faucet is placed before the drawer, making it convenient for the agent to move the handle to the right while opening the drawer. More detailed skill maps are shown in the Appendix.

## Ablation Study

**Importance of Code Reinitialization and Instruction Recovery:** Figure 3 shows that the mutual information between

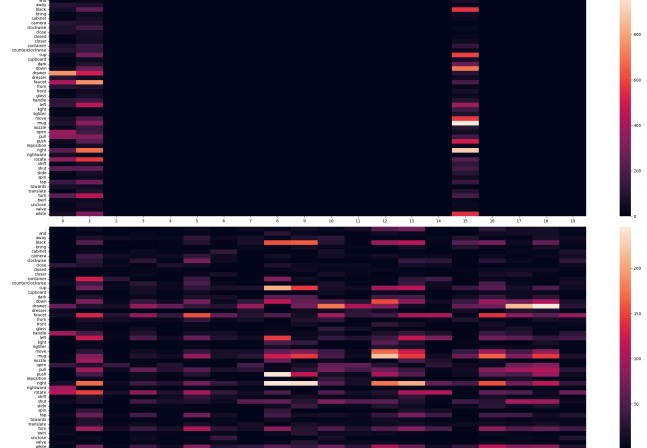

Figure 6: Skill-language mapping in LORel state environment. Up: skill-language graph on LISA(single encoder); Down: skill-language diagram of our LCSD.

language and skills is significantly increased when using language decoder and code reinitialization. To clarify the specific meaning of discrete skills, we plot corresponding word frequency on LCSD with and without code reinitialization and reconstruction in Figure 6. The comparison between the two figures clearly indicates that without the support of these two methods, the selection of code skills is limited to a small number, which is also observed in the DT(Chen et al. 2021) model, with even greater severity. The usage of code reinitialization provided a significant improvement in this case. In Appendix we display more skill maps with different LCSD settings in different environments.

**Generality of skill learning method:** Our skill-learning method can be extended to different models. In DT-based models, we observed index collapse in LORel environments, leading to poor test results. However, this problem was resolved by adding code reinitialization (Table 1). Figure 5

shows more detailed experiment results on LORel. By comparing our approach with LCSD combined with DT, it becomes evident that our approach can be used for the DT-based model for better skill discovery. More skill frequency figures and results are shown in the Appendix.

**Stability in multi task settings and varying parameters** We conducted detailed experiments on different types of settings in the LORel environment, as shown in Figure 5, to investigate whether language settings affect the model's performance in multi-tasks. By manipulating various words within sentences, we aimed to enhance skillful semantic comprehension. "seen tasks" refers to language descriptions that were identical to the training set. At the same time, "human" indicates completely different sentences that convey the same meaning. Our LCSD with code reinitialization and language recovery outperformed other methods in almost all the task settings. In VQ-VAE, the number of skill vectors in the codebook $M$ and the number of combined skill vectors in the language decoder are relatively essential parameters. However, we found our LCSD to be robust enough to these choices, as shown in the Appendix.

Table 3: **Inference time of LCSD and DT based model in three benchmarks**(second per episode).

| Policy | Timestep $N$ | CALVIN | LORel | BabyAI |
|---|---|---|---|---|
| DDPM | 25 | 297 | 623 | 560 |
| | 50 | 619 | 720 | 840 |
| | 75 | 880 | 1023 | 1240 |
| | 100 | 1200 | 1400 | 2000 |
| DDIM | - | 256 | 525 | 450 |
| DT | - | 225 | 801 | 500 |

**Inference time of Diffusion policy:** The diffusion policy's inference phase is time-consuming and depends on the hyperparameter timestep $n$. Therefore, we explored the impact of different timestep values in the BabyAI environment in the appendix. When set to 100 in the experiment, the evaluation time is approximately two to three times longer than the DT-based model. We list different inference times on different timestep among three benchmarks in Table 3. To ensure both efficiency and accuracy, we recommend that the n timestep should be defined as 50 for the experiment. For further study, we adopt the DDIM evaluation phase on our model and set the time step to 10. In this case, LGSA performed faster than DT during the evaluation phase and performed better than DDPM with low timesteps.

## Conclusion

In this paper, we have presented LCSD, a novel skill-based imitation learning framework designed for the purpose of multi-task skill discovery and behavior cloning in a language-conditioned environment. Our approach has demonstrated excellent performance in generating discrete skills while aligning with language in different environments. By initializing with diverse codes and establishing a stronger connection between skills and language through the language decoder, we have achieved more accurate and stable skill representations.

**Limitations and Future Work:** Our approach does not analyze the interconnections between different skills, which can be crucial in multi-tasking problems that are typically decomposed into a series of related sub-tasks. It is an interesting avenue for future research, with the potential to learn powerful skills to extend to more unknown tasks.

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
