# OpenReview forum: "Rethinking Mutual Information for Language Conditioned Skill Discovery on Imitation Learning"
_icaps-conference.org/ICAPS/2024/Conference — ICAPS 2024_

### Official Review · Reviewer_CaBx · 2024-01-16

**Significance And Importance:** 2
**Soundness:** 4
**Novelty:** 2
**Clarity:** 4
**Overall Evaluation:** 2
**Confidence:** 3

**Weaknesses:**

1: Minor weaknesses that are easily fixable.

**Contributions Of The Paper:**

The paper "Rethinking Mutual Information for Language Conditioned Skill Discovery on Imitation Learning" introduces the Language Conditioned Skill Discovery (LCSD) method. This method  employs mutual information theory for “skill discovery” in language-conditioned environments, particularly for robotic navigation and manipulation tasks. The mathematical contributions involve techniques in information theory, applied to correlate skills and language instructions. This research is important as it addresses the interpretability of skills in robotic tasks and aims to improve task success.

The key contributions of the paper include:

1. Two-Stage Imitation Learning Structure: Incorporating an encoder-decoder model for skill acquisition and a conditional diffusion policy for action prediction.
2. Mutual Information Skill Learning: Maximizing mutual information between language instructions and discrete skills, especially in multi-task environments.
3. Vector Quantization and Skill Sequence Analysis: Employing vector quantization for learning discrete latent skills and utilizing skill sequences from trajectories for semantic reconstruction of language instructions.
4. Evaluation in Diverse Scenarios: The paper demonstrates LCSD's effectiveness in language-conditioned robotic navigation and manipulation tasks, outperforming previous methods in various test environments.
5. Enhanced Generalization and Task Completion: LCSD shows improved generalization capabilities and higher task completion rates, indicating its potential for broad application in robotics and AI.

Details about the proposed framework

The proposed LCSD method bridges language instructions and discrete skills in robotic tasks using mutual information theory. Mutual information quantifies the shared information between a skill's representation and the original language instruction, enhancing the accuracy and effectiveness of action predictions. Its based on mainly the following elements.

- Encoder-Decoder Model: The method incorporates an encoder-decoder framework for skill acquisition. The encoder is designed to process and interpret language instructions, translating them into a set of discrete skills. These skills are represented in a latent space that captures the essence of the instructions in a form that is usable for the robot.

- Conditional Diffusion Policy: Following skill acquisition, a conditional diffusion policy is used for action prediction. This policy takes the discrete skills as input and generates a sequence of actions that the robot should perform. The diffusion policy is conditional because it depends on the specific skills that have been encoded from the language instructions.

- Role of Mutual Information: In this context, mutual information serves as a measure to quantify how much information the representation of a skill shares with the original language instruction. By maximizing this mutual information, the LCSD method ensures that the skills are as informative and representative of the instructions as possible, leading to more accurate and effective action predictions.

**Ethical Considerations:**

(1) Not Applicable: The paper does not have any ethical considerations to address

**Nomination For Best Paper:**

No

**Questions For Authors:**

Apart from the questions in the weakness sections I have a few questions from the authors regarding the research presented.

1. Could the authors elaborate on how the LCSD method performs in dynamic or unpredictable environments? Specifically, how does the method adapt to rapidly changing scenarios or cope with noisy language inputs and observations?

2. The paper doesn't delve into the proposed method's strategy for handling ambiguous or contradictory language instructions, which are common in real-world interactions. Could the authors provide more information on how the LCSD method interprets and responds to such instructions? Examples of particular interest include responding to subjective commands like 'move a bit to the left' or resolving contradictions in instructions, such as 'turn on the light but keep the room dark'.

3. Does the LCSD model support long-term learning and adaptation, where it can refine its skills and decision-making processes over time based on new data or experiences? How effectively can the model integrate continuous learning, and what are the challenges in implementing such a feature?

**Reproducibility:**

4: Authors promise to release code and domains (whichever apply).

**Strengths Of The Paper:**

1. The paper is well-written and features illustrative figures that aid in the comprehension (I specially like the illustration in Figure 2). The explanation of all the mathematical concepts seems correct to my understanding and not too overwhelming. I like that the authors have taken an extra step to validate that their method has more mutual information gain as compared to the baselines.(Figure 3)

2. A major strength of this paper is its use of mutual information to establish a link between discrete skills and language in imitation learning. This approach represents a theoretical advancement, introducing integration of information theory into the domain of imitation learning.

3. The focus on skill interpretability is also a strength. This aspect, often overlooked in similar research, is given due prominence here. The paper validates it through an ablation study and  experiments. These experiments demonstrate the method’s effectiveness in translating language instructions into robotic actions, enhancing the practical utility of the research.
     >> The use of skill codebooks and their visualization is an important aspect of this research. It provides a representation of how different language instructions correlate with specific robotic skills and actions. The paper demonstrates this through clear examples, such as how one skill in the codebook might be linked to turning an object to the right, while another relates to opening a drawer.

4. The evaluations conducted on multiple domains, comparing the LCSD method against state-of-the-art baselines like LISA and its adapted versions.

**Weaknesses Of The Paper:**

1. Limited Experimentation with Random Seeds:
    >> The experiments being run for only 3 random seeds may not sufficiently demonstrate the robustness and reliability of the LCSD method. In machine learning, especially in tasks involving randomness and stochastic processes, the choice of random seeds can significantly impact the results.
    >> The authors should consider running their experiments with more random seeds. Additionally, they could perform a statistical analysis to determine the significance of their results, thereby providing a more comprehensive understanding of the method's performance across different initial conditions.
    >> The authors can clarify this in their rebuttals.

2. Marginal Improvement Over Baseline Methods:
>>  The observation of only marginal improvement in task completion quality compared to LISA raises concerns about the practical significance of the LCSD method.
>> The authors can clarify this in their rebuttals.

3. Performance with Noisy Data and Language Instructions:
>> The uncertainty about how the LCSD method would perform with noisy data and language instructions is an important aspect. Robustness to noise is essential for practical applications, especially in real-world environments where noisy data is common.

4.  The paper should address the computational efficiency of their method, especially considering the integration of complex components like VQ-VAE and diffusion models. Is the increased computational cost justified by the improvements in performance? I see a marginal improvement in the success rates compared to LISA. Are there measures taken to address the sample complexity?
>> The authors can respond to this question in the rebuttals.

---

> ### Author Rebuttal · Authors · 2024-01-26
>
> Thank you for your comments and suggestions. We address your questions below:
> >Limited Experimentation with Random Seeds
>
> We chose three random seeds as the evaulation result because benchmark papers(MULC,LORel and LISA) we referenced show the results of three seeds. Actually we test five seeds for LCSD beyond baselines. We will explain this part in the final paper.
> >Marginal Improvement Over Baseline
>
> Our LCSD significantly outperforms LISA in two robotic environments. BabyAI is a two-dimensional grid game, where state and action spaces are simpler, making imitation learning sufficient to accomplish the tasks. LORel, as a complex robot environment for multilingual tasks, serves as the crucial benchmark for skill learning. In both LORel and CALVIN, LCSD demonstrates a clear advantage over LISA.
> >Computational Efficiency
>
> To reduce the computational cost of skill learning model, we employ simple MLP models for both the encoder and decoder.The increased skill training cost appears to be minimal. LCSD's computational time is mostly spent on imitation learning, with skill learning adding less than a 10% increase to the overall costs. More computational cost during the testing phrase arises in diffusion process, which we have detailed in Table 3.
> >Q1:Dynamic or unpredictable environment(Performance with Noisy data and language):
>
> We conducted experiments in LORel and CALVIN as noisy environments. In LORel, we perturbed language labels and presented a comprehensive comparison of the performance(Figure 5). We discussed CALVIN's unpredictability and difficulty in the appendix, and also demonstrated that skill learning methods in LCSD maintain stable performance amidst chaotic languages and sparse training sets.
> >Q2:Ambiguous or contradictory language:
>
> Good question! Language is complex and presents contradictions. As the second example you mentioned, LCSD tends to capture the action of turning on the light rather than comprehending the deeper intent of instruction, resulting in performing the action of turning on the light. Understanding more complex semantic information requires more advanced LLM.
> >Q3:long-term and new dataset:
>
> We perturb the language labels to assess the generalization capability of LCSD. While in CALVIN,  we conducted joint testing on four sequence tasks, with lengths of 120 steps to provide longer trajectory. We attempt to train LCSD on the ABC dataset and test on the D dataset, while still shows a 20% improvement compared to LISA.

---

### Official Review · Reviewer_Frvd · 2024-01-18

**Significance And Importance:** 1
**Soundness:** 2
**Novelty:** 2
**Clarity:** 3
**Confidence:** 2

**Weaknesses:**

0: Minor weaknesses requiring some work to be addressed for the paper to be accepted.

**Contributions Of The Paper:**

The paper presents a method in the context of language-conditioned policy learning. Authors propose an imitation learning approach named Language Conditioned Skill Discovery (LCSD).  The high-level goal is to enable robots to learn the connection between instructions in natural language and their perception and action capabilities. The approach uses a deep-learning architecture of two main components.  First, a (vector-quantized) variational auto-encoder (VQ-VAE) relates the current state and language instructions. Then, a diffusion policy with the U-net denoising model is used as the imitation policy.
The main contribution is the presentation of the method, which comprises a hierarchical skill learning Imitation policy, as described before. The evaluation showed that LCSD outperformed some other state-of-the-art methods in language-conditioned multi-task environments.  Another contribution is the examples of the interpretable discrete skills in different environmental conditions.

**Ethical Considerations:**

(1) Not Applicable: The paper does not have any ethical considerations to address

**Nomination For Best Paper:**

No

**Overall Evaluation:**

-1: (weak reject)

**Questions For Authors:**

- Which is the size of the language used in the robot instructions for the given benchmarks?

**Reproducibility:**

2: Some details are missing, but the paper still appears to be replicable with some effort.

**Strengths Of The Paper:**

The method considered the most recent state-of-the-art related work to address the common issues in imitation learning and to prepare the experiments.
The skill visualization shows an interesting result regarding the interpretability of how language and latent space are being related. The paper included supplemental material that provide additional evidence on their claims.

**Weaknesses Of The Paper:**

I’m not an expert in the most recent RL developments, nor in imitation learning. So, for me is difficult to assess the relevance of the contributions. From my point of view, the method combines a deep learning architecture that already has been used in the similar setting (VQ-VAE  for skill discovery in 2022) with a diffusion model, and adding complementary techniques such the skill reinitialization. Decisions in the design and the evaluation are sound, but since only the combination seems novel, I think the advance for the state of the art is limited.
On the other hand, the motivation of having unconstraint language instructions might justify the use of latent space representation. However, the setting from the evaluated environments (this is also true for related work) describes natural language with limited objects and actions that the robots can execute. So, I wonder how a simpler model would perform compared to these latent space plus quantize counterparts.
The paper assumes the reader has considerable prior knowledge regarding a specific setting within RL, which makes it difficult to read at the first time.  Additionally, the readability could also improve if concepts and elements are presented in the proper time. Just to give some examples.
- VQ-VAE was defined and referenced the third time it appears in the text.
- CLIP was referenced the third time it appears in the text.
- (Lines 240-254). The paragraph discusses results on mutual information when benchmarks or any experimental setting has not been presented.  It looks like it is misplaced here.

Other comments:
- Line 204. Why a1 is duplicated in the formula? This does not match the sample batch in Algorithm1
- The “skill learning” conditions in Algorithm 1 appears in the two periods and does not appear to be updated elsewhere. What is the intended idea of these if-statements.

---

> ### Author Rebuttal · Authors · 2024-01-26
>
> Thank you for your comments and suggestions. We address your comments and questions below:
> >Decisions in the design and the evaluation are sound, but since only the combination seems novel, I think the advance for the state of the art is limited.
>
> We provide a comprehensive theoretical analysis of mutual information in the context of multi-task language conditioned environment. Secondly, we implements skill learning method using an Encoder-Decoder architecture. Different from traditional VQ-VAE, the decoder is designed for better correspondence between languages and skills while donot directly participate in behavior cloning. Reinitialization method is introduced to alleviate index collapse,. Additionally, we introduce the Diffusion model as the imitation policy and design a multi-modal denoising model that supports various modalities. We are the first to establish the theoretical foundation of mutual information based on the analysis of the multi-task language conditioned environment, whereas previous studies focused solely on model design without providing a detailed analysis of skill interpretability.
> >the motivation of having unconstraint language instructions might justify the use of latent space representation. However, the setting from the evaluated environments describes natural language with limited objects and actions that the robots can execute. How a simpler model would perform compared to these latent space plus quantize counterparts.
>
> Language conditioned imitation learning treats language input as conditional information. We compared with these methods in Table 1. The 'Original' method involves language-based behavior cloning selected from different environments. 'Lang+DT' and 'Lang+Diffusion' are simpler imitation learning algorithms that exclude the skill learning part. Overall, our LCSD method significantly outperforms other simpler methods.
> >VQ-VAE was defined and referenced the third time it appears in the text. CLIP was referenced the third time it appears in the text. Line 204. Why a1 is duplicated in the formula?
>
> We apologize for these writing mistakes, and will further correct the errors in the final paper.
> >Misplaced results of mutual information
>
> The image in Lines 240-254 is placed here with the intention to provide more intuitive experimental support after the theoretical proposal. We apologize for any confusion this redundancy may have caused and will move this image to the experiment section in the final version.

---

### Official Review · Reviewer_DTRv · 2024-01-21

**Significance And Importance:** 2
**Soundness:** 3
**Novelty:** 3
**Clarity:** 4
**Overall Evaluation:** 2
**Confidence:** 2

**Weaknesses:**

1: Minor weaknesses that are easily fixable.

**Contributions Of The Paper:**

This paper introduces a novel approach for imitation learning based on vector quantization to learn discrete latent skills and leverage skill sequences of trajectories to reconstruct high-level semantic instructions.

**Ethical Considerations:**

(1) Not Applicable: The paper does not have any ethical considerations to address

**Nomination For Best Paper:**

No

**Questions For Authors:**

-

**Reproducibility:**

2: Some details are missing, but the paper still appears to be replicable with some effort.

**Strengths Of The Paper:**

The paper provides a detailed explanation of the new approach.

I found the experiments section quite complete. The authors show a comparison with several baseline techniques in three different benchmarks with different levels of difficulty. Moreover, they show a budge of analysis that provides insights about the scope of this approach.

**Weaknesses Of The Paper:**

-

---

> ### Author Rebuttal · Authors · 2024-01-26
>
> Thank you for reviewing our paper and for the positive feedback. We are pleased that you find our proposed method convincing and appreciate your recognition of our comprehensive experiments.

---

### Meta-Review · Area_Chair_8NEd · 2024-02-06

**Recommendation:** Accept (Poster)
**Confidence:** 3

**Metareview:**

This paper presented a multi-modal vision-language system for robotic skill acquisition.
It demonstrated that a loss function that is based on mutual-information is critical for effective acquisition.
MI was used in two ways, the forward form for I(z;l) and the reverse form for I(z;l | s).

While initially the scores were divided, the reviewer with a negative score was satisfied with the rebuttal and
raised the score to weak accept, resulting in 2-1-2 score, leaning toward acceptance.

The authors are encouraged to address the points raised by the reviewer before the camera-ready.

minor comments:

Line 205: "l" in the trajectory is not defined at this point.

In Table 1,

1. isn't the table caption supposed to be at the bottom of the table?
2. the "performance" needs to be briefly explained. What are these numbers?

Line 380: introduce BabyAI/Lorel/Calvin in the order it is presented in Table 1.

I believe the presentation in this paper has some room for improvement.
For example,
the figure captions only explain what is in the figure, and does not explain how to interpret the figure.
Looking at the caption of Figure 5,
I cannot tell what it is trying to tell until I look for the corresponding base text in line 498.
There is no reason for this bar graph to be a graph: It can be simply a table.
I agree that the proposed approach has a general upward trend compared to other algorithms.

**Ethical Considerations:**

(1) Not Applicable: The paper does not have any ethical considerations to address